

# Change and stasis of distinct sediment microbiomes across Port Everglades Inlet (PEI) and the adjacent coral reefs

Lauren E. Krausfeldt[1,*], Jose Victor Lopez[1,*],
Catherine Margaret Bilodeau[1], Hyo Won Lee[2] and Shelby L. Casali[3]

[1] Halmos College of Arts and Sciences, Nova Southeastern University, Dania Beach, Florida, United States
[2] Florida International University, Miami, Florida, USA
[3] Coral Reef Conservation Program (CRCP), Florida Department of Environmental Protection, West Palm Beach, Florida, USA
* These authors contributed equally to this work.

Corresponding author
Jose Victor Lopez, joslo@nova.edu

## ABSTRACT

Deep water ports are human built coastal structures that by definition welcome ship traffic and disturbance. Evidence is accumulating that enhanced port activities such as dredging or deepening have negatively affected nearby natural habitats. Port Everglades Inlet (PEI) is a large active South Florida cargo port for over two million people and lies adjacent to coral reefs, dwindling mangroves, and recreational beaches. In this study, the microbial communities of PEI and adjacent reef sediments were characterized to serve as indicators for change due to dredging and assess anthropogenic influence on these sensitive ecosystems by sequencing the V4 region of 16S rRNA ahead of a large-scale port deepening event. For the first time, this study established baseline bacterial community characterizations and their patterns of diversity prior to and after a maintenance dredging event. PEI samples were collected for two consecutive years 2020 (Phase I, before maintenance dredging) and 2021 (Phase II, after maintenance dredging) from PEI sediments and adjacent coral reef sediments. In spite of their proximity and tidal connections through the PEI, reef and PEI sediment microbial communities were distinct. Changes in microbial diversity within the intracoastal waterway (ICW), a route for community exchange or transfers, were the greatest after maintenance dredging occurred. Microbial diversity in reef sediments also changed after dredging, indicating potential influence from resuspended sediments due to an associated increase in trace metals and decrease in cyanobacterial diversity. Sediments were identified as a possible source of human and coral pathogens, although dredging did not affect the relative abundances of these indicator microorganisms. This study highlighted the utility and relative ease of applying current molecular ecology methods to address macroscale questions with environmental management ramifications.

## INTRODUCTION

Port Everglades Inlet (PEI) is located on the East coast of Florida, U.S.A situated in three municipalities: Fort Lauderdale, Dania Beach, and Hollywood (https://www.porteverglades.net/). PEI has been in operation for almost a century, harboring cargo ships in 1928 and federally authorized for passenger ships in 1930. Since then, this highly engineered deep water port is 1,923 feet (641 m) in length by 885 feet (295 m) wide with a depth of 39 feet (13 m, Stauble, 1993). The PEI and adjacent intracoastal waterway (ICW) are subject to a high volume of commercial and recreational boat traffic and likely influenced by local and regional human-derived (*e.g.*, "built") anthropogenic inputs through a myriad of urban land uses well beyond its boundaries into its associated watershed. Within the next few years, the PEI will soon be one port looking to expand traffic by deepening and widening. This follows other developments across the globe to accommodate Neo-Panamax ships that were added to the fleet after the expansion of the Panama Canal in 2016 (*Ashe, 2018*). Port Everglades Deepening Project (PEDP) received federal authorization in December of 2016 for the U.S. Army Corps of Engineers to move forward with the deepening and widening of PEI channels as part of the Water Infrastructure Improvements for the Nation (WIIN) Act (https://www.usace.army). The Army Corp of Engineers released an Environmental Impact Statement (EIS) in late 2020 for public comment describing the possible effects of deep dredging from the current mean 42 feet (12.8 m) to a depth of 48 feet (14.6 m) in PEI (*Army Corps of Engineers, 2020*).

Urban growth and port development can have detrimental effects on surrounding natural habitats, including coral reef systems (*Walker et al., 2012*). The S. Florida greater metropolitan area currently holds a population >6 million people (U.S. Census Bureau as of April 2020) and continues to grow and exert pressures on proximal natural habitats. Several studies have shown that resuspended sediments and accompanying increased turbidity of seawater can degrade coral reef health around the world (*Dodge & Vaisnys, 1977*; *Fabricius, 2005*; *Wolanski, Martinez & Richmond, 2009*; *Bessell-Browne et al., 2017*; *Nascimento et al., 2020*). To protect vital and sensitive coastal habitats, the specific chemical and microbial threats and their origins should be accurately located, characterized, and identified.

Microbial communities provide important ecological and biogeochemical services, therefore microbial community profiling reveals valuable ecosystem information (*Thompson et al., 2017*; *Egger et al., 2018*). When combined with corresponding environmental metadata, microbes can provide sourcing information about water masses and serve as indicators of degradation or alteration of water quality. Microbes also act as integral symbionts to most resident organisms, such as on sensitive ecosystems like coral reefs. For example, *Peixoto et al. (2021)* have shown certain microbial symbionts positively affect and protect coral species. *O'Connell et al. (2018)* provides a weekly sampling of PEI surface waters and found a fairly stable microbial composition, with increased microbial abundance and richness in the early spring and late summer months, most likely caused by increased temperatures, UV radiation, and precipitation. Thus, understanding microbial dynamics positively impacts the health of human and resident marine life.

In this context, we aimed to characterize the microbial communities in the sediments of the PEI and the proximal coral reef before and after routine operational and maintenance (O&M) dredging events in January and March 2021. The baseline profiling was used to determine the resident microbiomes and microbial diversity of the PEI and reef sediments, and the changes that came after O&M dredging served as indicators of the potential influence of these activities. Microbiome information can be used by environmental managers to provide valuable context on the adverse effects that dredging on built environments with heavy anthropogenic influence may have on adjacent marine natural habitats (*Miller et al., 2016*; *Cunning et al., 2019*; *Niu et al., 2021*).

# MATERIALS AND METHODS

## Sampling sites and metadata collection

For this study, a total of 240 surface marine sediment samples from the top 10 cm were collected (120 per year in 2020 and 2021) from 40 sites spread over PEI (or port) and the adjacent Florida coral reef habitat to characterize the microbial community of sediment in these areas (Table 1; Fig. 1). To characterize the chemical and physical composition of the water and sediment supporting that community, a subset of nine sediment samples and three water samples (split into multiple bottles) per year were also analyzed. Water and sediment samples were delivered on ice to the CAChE Nutrient Analysis Core Facility at Florida International University for chemical analyses of trace metals and minerals, Total Phosphorus (TP), Total Nitrogen (TN), Total Organic Carbon (TOC) and Total Carbon (TC) (Table S1). The sample IDs reflect the buoy collection site shown in Fig. 1 where three replicates per site were collected along a transect (−00, 15, or 30 m from the buoy). At sites R11, P09, and P16, water samples were taken from mid-depth for the water analytes (Tables 1 and S1, Fig. 1). At P02, P09, P16, R01, R09, R11, R12, R15, and R22, sediment samples were collected in the middle of the transect for the sediment analytes (Table 1 and S1, Fig. 1).

## Microbiome analysis

All 240 sediment samples were processed for DNA analyses. Preparation of DNA samples from sediment and water acquired for high-throughput 16S rRNA gene sequencing were conducted as described in detail in previous studies (*Campbell et al., 2015*; *O'Connell et al., 2018*; *Easson & Lopez, 2019*). Briefly, after pelleting 1–2 grams of sediment, genomic DNA was extracted using the DNeasy PowerLyzer Powersoil kit (12855; Qiagen, Hilden, Germany) following standard protocols. DNA quality was evaluated for molecular weight integrity by electrophoresis on a 1.0% agarose gel.

Microbial 16S rRNA gene libraries were generated by amplifying the V4 region with universal primers 515F and 806R with the Golay barcodes and Illumina adapters attached to the reverse primer following the protocol recommended by the Earth Microbiome Project (*Caporaso et al., 2012*; *Thompson et al., 2017*). These primers have been selected because they can amplify and provide the most comprehensive diversity of both Bacteria and Archaea. The resulting 16S rRNA gene libraries were purified with AMPure beads and prepared for sequencing on an Illumina MiSeq following the manufacturer's protocol for

**Table 1 List of all sample sites and coordinates.** Each site had three replicates taken for a total of $N = 120$ samples in 2020 and 2021. R, Reef; P, Port (PEI).

| Sample number | Latitude | Longitude | Location name | Nutrient analysis |
|---|---|---|---|---|
| R01 | 26.11233 | −80.09223 | 2000N-IRL-Control | Sediment |
| R02 | 26.10795 | −80.09242 | 1500N-IRL | |
| R03 | 26.09893 | −80.09682 | 500N-CPS | |
| R04 | 26.09893 | −80.09320 | 500N-IRL | |
| R05 | 26.10070 | −80.08772 | 700N-MRL | |
| R06 | 26.10070 | −80.08320 | 700N-ORL | |
| R07 | 26.09713 | −80.08780 | 300N-MRL | |
| R08 | 26.09713 | −80.08352 | 300N-ORL | |
| R09 | 26.09545 | −80.10138 | 50N-NRC | Sediment |
| R10 | 26.09545 | −80.09462 | 100N-IRL | |
| R11 | 26.09587 | −80.08882 | 150N-MRL | Sediment and water |
| R12 | 26.09512 | −80.08400 | 50N-ORL | Sediment |
| R13 | 26.09193 | −80.10272 | 100S-NRC-RS | |
| R14 | 26.09193 | −80.09365 | 100S-IRL | |
| R15 | 26.09193 | −80.08888 | 100S-MRL | Sediment |
| R16 | 26.09240 | −80.08418 | 50S-ORL | |
| R17 | 26.08988 | −80.09945 | 300S-NRC-CPS | |
| R18 | 26.08988 | −80.09382 | 300S-IRL | |
| R19 | 26.08988 | −80.09125 | 300S-MRL | |
| R20 | 26.08988 | −80.08413 | 300S-ORL | |
| R21 | 26.07895 | −80.09447 | 1500S-IRL | |
| R22 | 26.07447 | −80.09548 | 2000S-IRL-Control | Sediment |
| P01 | 26.11111 | −80.11028 | Control-N | |
| P02 | 26.09833 | −80.11806 | NTB-1 | Sediment |
| P03 | 26.09444 | −80.11750 | NTB-2 | |
| P04 | 26.09389 | −80.11222 | EC-1 | |
| P05 | 26.09417 | −80.10694 | EC-2 | |
| P06 | 26.09222 | −80.11972 | MTB-1 | |
| P07 | 26.09139 | −80.11306 | MTB-2 | |
| P08 | 26.09278 | −80.10833 | EC-3 | |
| P09 | 26.09028 | −80.11583 | MTB-3 | Sediment and water |
| P10 | 26.08750 | −80.11861 | STB-1 | |
| P11 | 26.08639 | −80.11806 | STB-2 | |
| P12 | 26.08583 | −80.11278 | SAC-1 | |
| P13 | 26.08250 | −80.11333 | SAC-2 | |
| P14 | 26.07806 | −80.11417 | SAC-3 | |
| P15 | 26.07444 | −80.11778 | STN-1 | |
| P16 | 26.07306 | −80.11667 | STN-2 | Sediment and water |
| P17 | 26.06583 | −80.11472 | SAC-4 | |
| P18 | 26.04833 | −80.11639 | Control-S | |

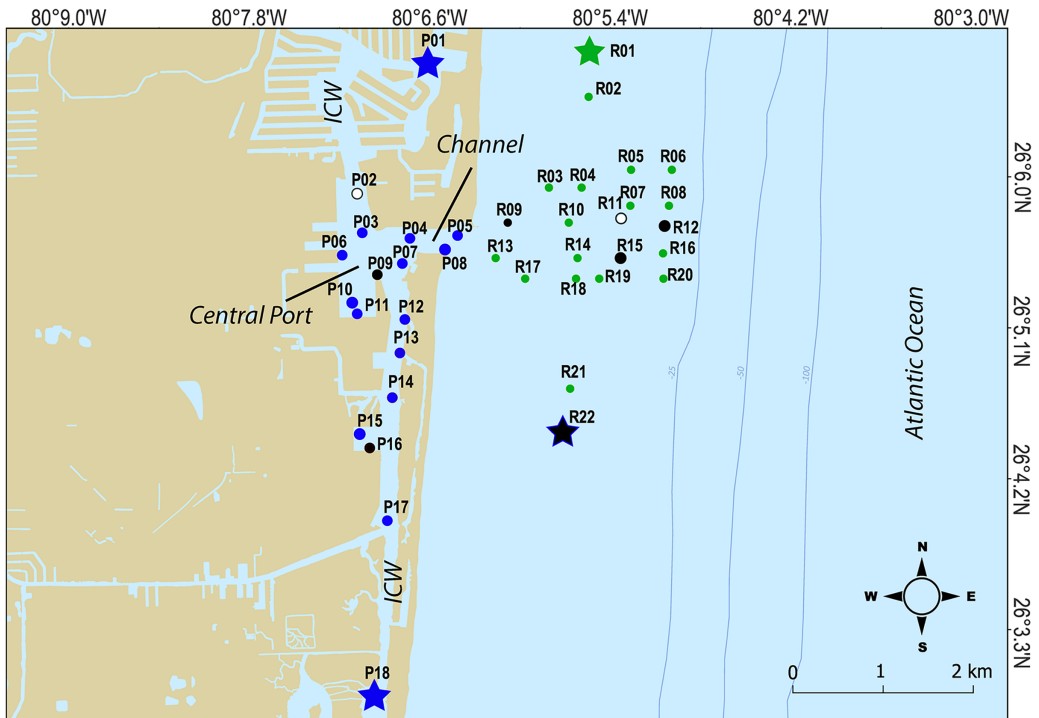

**Figure 1 Map of sampling sites.** Sites where chemical data was collected from sediments are indicated as black closed circles, while sites where chemical data was collected from the water column are indicated by white circles. Hopper dredging occured from Feb 11 to Mar 5 2021. Sites that served as "no dredging" controls are indicated by stars. P indicates sites where samples were collected that represent the Port Everglades Inlet, and R indicates sites where sediment was collected from the adjacent coral reef.

library preparation and Miseq Reagent Kit v3 for 250 bp paired end reads and with custom indices needed for protocol outlined by the Earth Microbiome Project (*Thompson et al., 2017*; *O'Connell et al., 2018*; *Easson & Lopez, 2019*).

Raw data was demultiplexed using demux in the Qiime2 environment (*Bolyen et al., 2019*). Resulting FASTQ DNA sequence files were uploaded to the CosmosID-HUB where sequences were trimmed using bbduk (minlen = 40 qtrim = rl trimq = 25 maq = 30) and clustered at 97% to generate operational taxonomic units (OTUs). Relative abundances, taxonomic annotation, and Chao1 richness calculations were performed in the CosmosID-HUB. Statistically significant differences in Chao1 richness between groups were determined using the Wilcoxon Rank Sum Tests with a *p* value cut off of 0.05. Results were confirmed on square root transformed and untransformed data outside of CosmosID-HUB using function specpool as a part of the vegan package in R. The relative abundances of OTUs from CosmosID-HUB were used for beta diversity analyses, including SIMPER, ANOSIM, and non-metric multidimensional scaling (NMDS), which were performed using Primer-E v7 (*Oksanen et al., 2017*). Averages were calculated per site using replicates for these analyses. Using replicate samples as individual samples did not change the results.

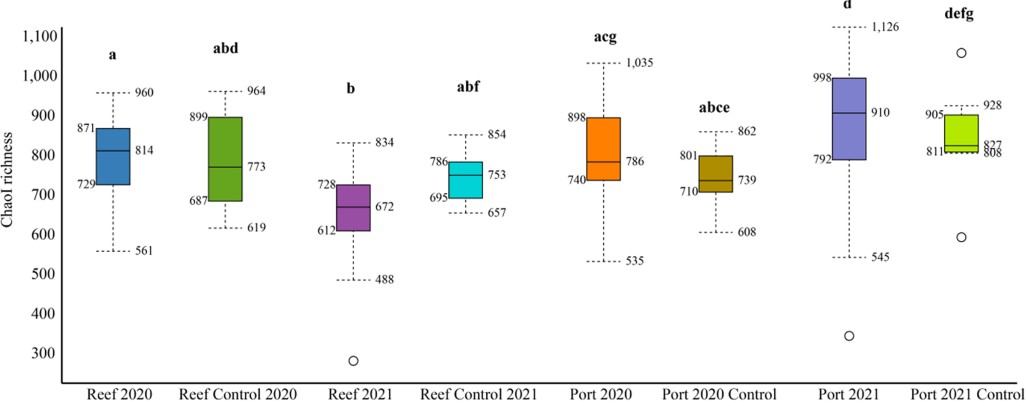

**Figure 2 Alpha diversity box plots depicting Chao1 species richness for 2020 and 2021 PEI and reef sediments.** Statistically significant differences, represented by the letters, in Chao1 richness measurements were determined using the Wilcoxon Rank Sum Tests.

Metadata is available at our NSUworks repository: https://nsuworks.nova.edu/lopez_lab/3/. Sequence data has been submitted for public access to the National Center of Biotechnology Information (NCBI) Sequence Read Archive (SRA). The NCBI SRA accession is PRJNA742832, release date: 2022-04-01.

## RESULTS

### Analyses of 2020 and 2021 R and P sediment microbiomes

After QC and sequence processing, a total of 230 16S rRNA amplicon libraries were sequenced and analyzed from the PEI and the adjacent coral reef. The final 2020 dataset included 117 total samples (63 reef and 54 PEI samples), while the final 2021 dataset included 113 samples (51 PEI and 62 reef samples). All 230 samples had a sequence quality level of Q30 or above. Sequence results from 10 samples were omitted from further analyses when the read number fell below 50,000.

The mean number of 16S rRNA reads per sample was 86,722 and 124,663, in 2020 and 2021, respectively (Table S2). Across all datasets, a total of 2,620 genera and 1,190 families were identified. The mean alpha diversity (Chao1 richness) for the PEI sediments were 780 and 910 OTUs in 2020 and 2021, respectively, and 814 and 693 in 2020 and 2021 in the reef sediment, respectively (Fig. 2). There was no difference in Chao1 richness in 2020 between the PEI and reef sediments, but in 2021, the sediments of the PEI were more enriched in microbial OTUs compared to the PEI in 2020 ($p = 0.01$) and the reef in 2021 ($p = 0.001$, Fig. 2). The Chao1 richness in the reef sediments decreased in 2021. Chao1 richness in samples that served as undredged controls did not change from 2020 to 2021 in the PEI or the reef (Fig. 2).

NMDS analyses were used to evaluate beta diversity *via* Bray-Curtis similarity between microbial communities at the family level which revealed at least two distinct clusters separating the PEI and reef sediments when evaluating both years together (Figs. 3A and 3B, ANOSIM, $R = 0.726$, $p = 0.001$). Sites representing the channel (P04, P05, P07, and

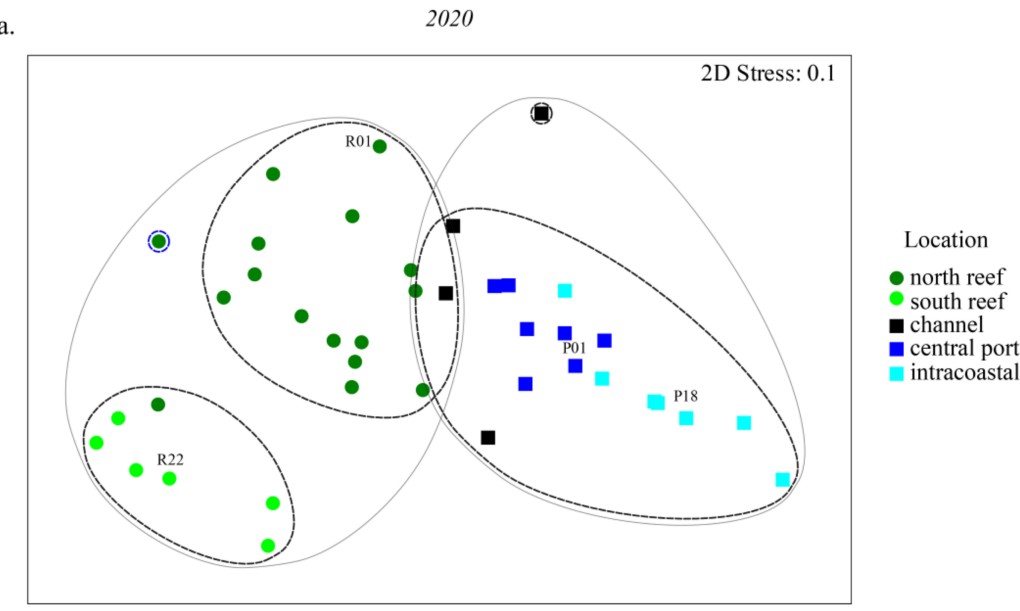

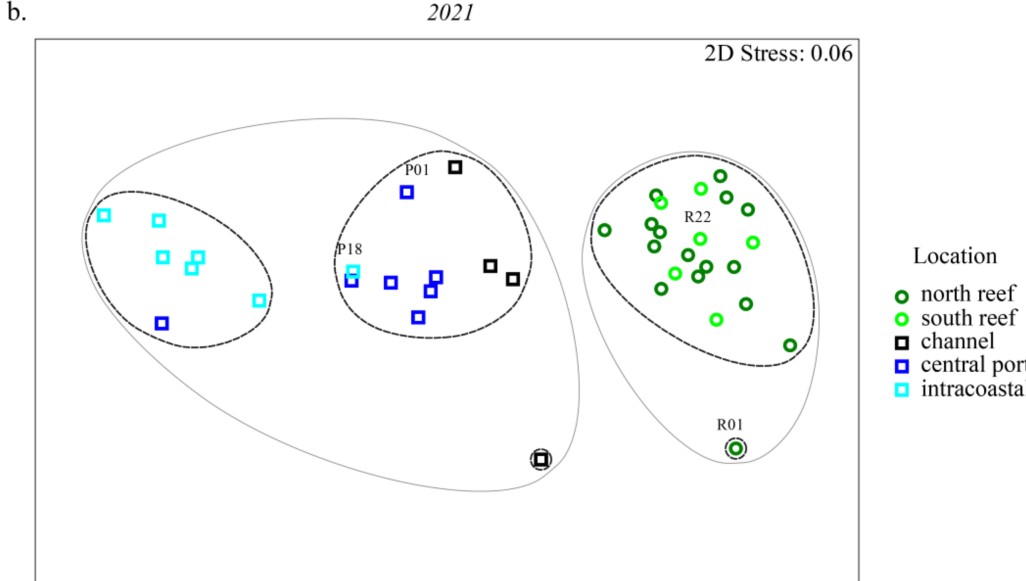

**Figure 3 Differences in microbial community composition between sediment samples from the reef and PEI across 2 years.** NMDS analyses showing relationships between sediment samples from reef and PEI based on beta diversity of microbial communities calculating Bray-Curtis similarity after square root transformation of relative abundances from 2020 (A) and 2021 (B). Differences between subsections of the reef between years were determined using ANOSIM. Cluster analysis was performed at 75 (gray solid line) and 80 (black dashed line) percent. Control (non-dredged) sites are labeled with site name. Symbols representing 2020 are closed symbols and 2021 are open symbols. Green circles, north reef; light green circles, south reef; black, channel; dark blue, central port (*e.g.*, turning basin); light blue, intracoastal waterway.

P08) separated these groups. Reef sediments in 2020 clustered into two distinct groups that represented north (R01–R16) and south reef sites (R17–R22), which included control samples (sites R01 and R22; ANOSIM, R = 0.830, *p* = 0.001). This distinct clustering of the

north and south reef sites was not observed in 2021 (ANOSIM, R = −0.1, $p$ = 82.9) but there was a stronger difference between the reef and PEI sediments in 2021 (ANOSIM, R = 0.863, $p$ = 0.001). Indeed, differences between the south reef sediments in 2020 and 2021 were strong (ANOSIM, R = 1, $p$ = 0.001). The south reef sediments, including the control (R22), clustered with the north reef sediments in 2021 (Fig. S1).

The differences between the PEI sediments in 2020 and 2021 were weak but significant (ANOSIM, R = 0.216, $p$ = 0.002). These differences were due to the changes in microbial diversity in the ICW (sites P12–P18) from 2020 to 2021, which clustered distinctly from each other (ANOSIM, R = 0.525, $p$ = 0.011, Fig. S1). In 2021, there were two distinct clusters amongst samples from PEI sediments that distinguished the central PEI (*e.g.*, turning basin, sites P01–P11, ANOSIM R = 0.501, $p$ = 0.009) and channel (ANOSIM, R = 0.921, $p$ = 0.003) from samples collected in the ICW. The control samples in the PEI, P01 and P18 where there was no dredging, still clustered with the PEI samples as they did in 2020.

In both 2020 and 2021, the most abundant phylum across sites was Proteobacteria, which on average made up 56 ± 6% of the microbial community (Fig. 4A) and was comprised of classes Gammaproteobacteria (30.8 ± 6.0%), Deltaproteobacteria (15.0 ± 6.1%), and Alphaproteobacteria (9.5 ± 3.5%), with the most abundant orders including the Desulfobacteriales (7.6 ± 4.8%), E01-9c-01 marine group (7.0 ± 2.0%), Xanthomonadales (3.9 ± 1.3%), Cellvibrionales (3.4 ± 1.3%), Rhodospirillales (2.9 ± 1.6%), Rhodobacterales (2.4 ± 1.6%), Steroidobacterales (2.3 ± 0.7%), Rhizobiales (2.0 ± 0.7%), Myxococcales (1.6 ± 0.6%), and Thiotrichales (1.5 ± 1.2%, Figs. 4B and S2). Other dominant phyla included Planctomycetes (7.4 ± 2.0%), Bacteroidetes (6.7 ± 2.2 %), and Cyanobacteria (5.3 ± 4.8%) representing other more abundant orders Planctomycetales (3.6 ± 1.4%), uncharacterized Oxyphotobacteria (3.4 ± 2.6%), Sphingobacteriales (1.6 ± 1.0%), Pirellulales (1.4 ± 0.6%), Cytophagales (1.3 ± 0.5%), and Flavobacteriales (1.1 ± 0.7%, Fig. 4B). Another dominant order across all samples included Nitrosopumilales (3.2 ± 2.1%) of the phylum Thaumarchaeota.

The average similarity at the family level between PEI and reef sediments considering both years was 72.6% (SIMPER). Families from multiple phyla, including Desulfobacteraceae, Oxyphotobacteria, Desulfobulbaceae, Rhodospirillaceae, uncharacterized Gammaproteobacteria, Anaerolineaceae, and Thiotrichaceae, accounted for the top ~10% of the dissimilarities (SIMPER). Abundances of Anaerolineaceae, Thiotrichaceae, Desulfobacteraceae, and Desulfobulbaceae were higher in the PEI sediments and Oxyphotobacteria and Rhodospirillaceae were more abundant in the reef sediments. Cyanobacterial abundances decreased in the reef sediments in 2021, driving differences in microbial diversity between years at this location, especially in the south reef sites (SIMPER). Differences in the intracoastal in 2020 *vs* 2021 were driven by increases in the relative abundances of Archaea and Chloroflexi, the total of which increased from 3.5 ± 2.2% to 10.2 ± 5.6% and 6.3 ± 3.3% to 8.7 ± 4.6% respectively, which included the orders Thermoplasmatales, Crenarchaeota, Asgardeota, Lokiarchaeota, Dehalococcoidia, and Anaerolinae (SIMPER). This corresponded to the decrease in the relative abundances of several families of Proteobacteria from 61.3 ± 5.4% to 50.4 ± 9.6%.

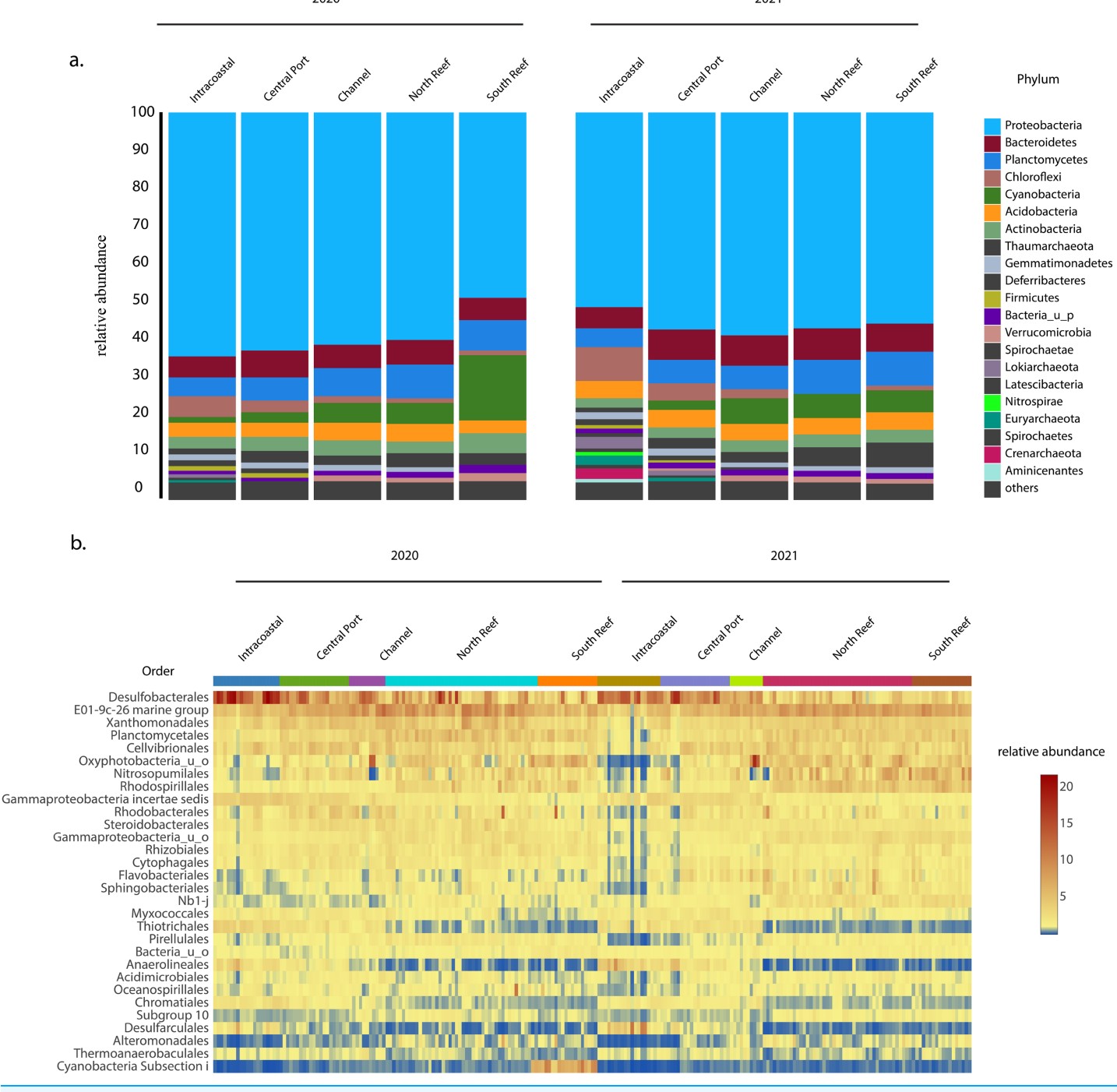

**Figure 4 Most abundant taxa in the reef and PEI sediments in 2020 and 2021.** Most abundant taxa in the reef and PEI sediments in 2020 and 2021 represented by the phyla that made up 95% of the community (A) and the 30 most abundant orders in the PEI and reef in 2020 and 2021 (B). Subsections within the reef and PEI reflect designations in previous Figs. 1 and 3.

Potential pathogens associated with Stony Coral Tissue Loss Disease (SCTLD) were identified, specifically from some of the most dominant orders observed in this study Rhizobiales and Rhodobacterales which were present in the sediments of both the PEI and reef before and after dredging (Figs. 4B and S3). The most abundant families of Rhizobiales

in the reef samples were Hyphomicrobiaceae (0.6 ± 0.3%), Rhodobiaceae (0.3 ± 0.1%), Rhizobiaceae (0.3 ± 0.2%), Methyloligellaceae (0.2 ± 0.1%), and Ocs116 clade (0.1 ± 0.1%). Rhodobacteraceae was the highest in relative abundance potential coral pathogen in the order Rhodobacterales at 2.4 ± 1.8%. There was no evidence that these orders increased after dredging in the reef sediments (Kruskal-Wallis, $p = 0.10$).

Some potential human pathogens and enteric taxa were observed, although their relative abundances were low (<0.05%). These include *Staphylococcus, Streptococcus, Enterococcus aerogenes*, and *Vibrio*. *E. aerogenes* (although in low relative abundances of <1%) was ubiquitous in PEI and reef sediments. The results reflect recent surveys of waterborne bacterioplankton from same Broward County area (*Campbell et al., 2015*; *O'Connell et al., 2018*).

TN, TP, TC, and TOC and trace metals and elements were measured in a subset of samples to represent the chemical profiles of sediments in the PEI and reef (Figs. S4A and S4B). The samples clustered distinctly by PEI and reef sediments when considering these variables alone (Fig. 5). TP and several trace metals, including Al, Fe, Co, Cu, Zn, Mo, Pb and the element Se were higher in the PEI compared to the reef sediments in both years (Table S3). Only Se was higher in the reef in 2021 compared to the reef sediments in 2020 (Table S3; Fig. S4).

Several of these variables correlated to the differences in microbial communities (Fig. 5B). Se and TP correlated to bacterial assemblages when considering all sediment samples in the PEI and reef (BEST, $R^2 = 0.533$, $p = 0.001$). However, the bacterial assemblage of the sediments of the PEI and reef in 2020 alone correlated to Sb and TP (BEST, $R^2 = 0.530$, $p = 0.001$), and in 2021, they strongly correlated to Be, Mo, and Pb (BEST, $R^2 = 0.813$, $p = 0.001$). Pb also strongly and positively ($R^2 > 0.90$) correlated to Hg, Zn, Cu, Ni, Co, and Al and therefore Pb was used as a proxy for this group of trace metals and minerals to correct for variation inflation.

## DISCUSSION

### Characterization of the sediment microbial communities in built (PEI) and adjacent natural environment (reef)

As one of five inlets in the three county (Palm Beach, Broward and Miami-Dade) metropolis of S. Florida (U.S.A), the PEI, a commercial port, serves as a major connection between the ICW and the shallow coral reefs of S. Florida. PEI also lies adjacent to additional diverse habitat types–a depauperate mangrove preserve, urban and residential canals, and recreational beaches. This study (first sanctioned as Project CRCP 13 by DEP CRCP—https://floridadep.gov/rcp/coral/documents/analysis-sediments-port-everglades-inlet-pei-microbiome-characterization-phase) has provided an opportunity to characterize marine sediments from a heavily built (PEI) environment closely juxtaposed to a more natural (reef) and sensitive habitat.

High alpha diversity was not surprising in the PEI and reef sediments in 2020, which served a baseline before dredging, as sediments are known to hold high species richness compared to other types of habitats (*Delgado-Baquerizo et al., 2018*). However, analyses

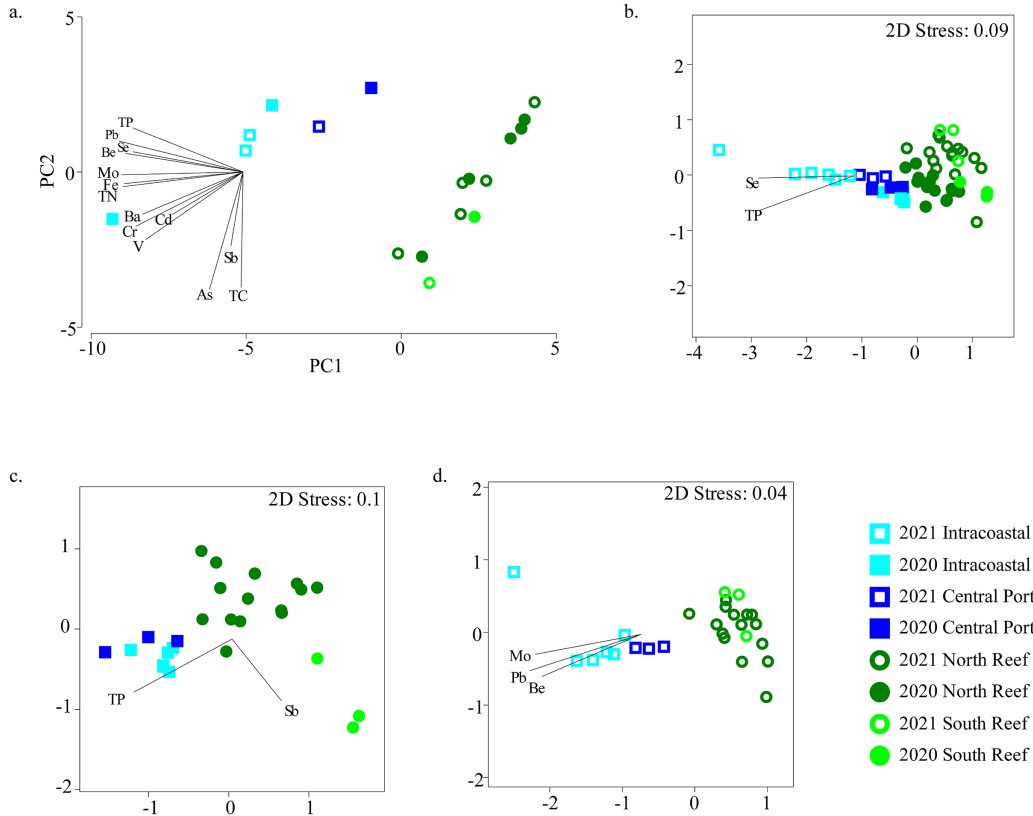

**Figure 5 Relationships between microbial community diversity and chemistry of the reef and PEI sediments.** Principal coordinates analysis was used to assess relationships between samples based on environmental variables measured in the sediments in the PEI and reef (A). Non-metric multi-dimensional scaling analyses using Bray-Curtis similarity after square root transformation of relative abundances were performed to examine beta diversity of microbial communities of the reef and PEI sediments that had corresponding environmental data for 2020 and 2021 (B), 2020 alone (C), and 2021 alone (D). Overlaid vectors in B–D represent environmental variables that correlated (Pearson $R^2$) significantly to the beta diversity of microbial communities determined by BEST analysis.

using beta diversity to compare the two locations demonstrated that the microbial communities in the PEI and reef sediments were distinct in their composition (Fig. 3). This was regardless of previous dredging activity, and the microbial community assemblages in the PEI became even more distinct from the reef sediments in microbial community composition in 2021 and became richer in microbial OTUs (Figs. 2 and S2). The PEI sediments were more enriched in Anaerolineaceae, which are chemoheterotrophic anaerobes and known for their ability to degrade alkanes, Desulfobacteraceae and Desulfobulbaceae, which are also strict anaerobes and reduce sulfate, and Thiotrichaeacea, which are sulfur oxidizers (*Wasmund, Mußmann & Loy, 2017*). These observations align with high sulfate concentrations observed in upper marine sediments and previous literature describing the importance of sulfur cycling to other biogeochemical cycles in marine environments (*Jørgensen, Findlay & Pellerin, 2019*). Compared to the PEI, the reef sediments had higher relative abundances of cyanobacteria, especially in 2020 in the south reef sediments (Figs. 4A and 4B), indicating there was likely more light penetration to the

reef sediments compared the PEI. Indeed, cyanobacteria have been shown to be important in N-cycling and calcification in coral reef sediments (*Werner et al., 2008*). These results highlight the potential differences in biogeochemical cycling that occurs in each of these sediments and suggest that introduction of sediments from PEI to the reef could have impacts on greater ecosystem functioning.

Although no major hurricanes hit S. Florida in 2020–2021, more dynamic wave action and stronger ocean currents are expected offshore than within the PEI and ICW, which could be responsible for some community restructuring observed in the reef between years (Fig. 3). Regarding how marine sediment microbiomes could shift and change, knowing the patterns of water body currents within the ICW would be helpful but are not available at this time. The PEI is not devoid of water currents and dynamics. Of course, flow within the PEI channel itself will be stronger than wider basins, especially during tidal surges. However, it is known that in the PEI, the average tidal current speed is 0.7 knots and can reach speeds up to 3 knots at flood tide and 5 knots at ebb tide. Prominent winds that blow from the southeast and east also travel at speeds of 17 knots or greater, causing very strong currents at the entrance to the inlet (*National Ocean Service, 2021*). Sediments can be easily disturbed through normal PEI shipping activities and without active dredging events. For example, the samples for this study were collected from the top 10 cm of the sediment layer. Large tanker ships regularly stir up sediments in the turning basin of the PEI.

## Effects of dredging of the PEI on sediment microbial communities

Sampling was conducted to address possible impacts of dredging of the PEI to the reef. The microbial communities of PEI sediments shifted into two distinct clusters in 2021 which did not exist in 2020 data (Figs. 3 and S2), indicating the microbial communities of the ICW were more distinct after dredging. Although this could be perhaps be attributed to increased boat and ship traffic stirring up sediments, the undredged control sites within the ICW remained clustered with the rest of the central PEI samples from 2020 and channel samples, which were dredged. This indicated that dredging had a larger impact on the ICW sediments compared to the central PEI (*e.g.*, turning basin) and the channel. Those changes in microbial community assemblages were associated with increased abundances of Chloroflexi and several phyla of Archaea and a decrease in Proteobacteria, which may be reflective of microbial community assemblages in the deeper marine sediments of the PEI. Indeed, Chloroflexi and many Archaea are obligate anaerobes and prevalent in anaerobic conditions that would be expected there. Their increase after dredging may have implications on biogeochemical cycles in the upper marine sediments in the PEI, as Archaea and Chloroflexi are important players in carbon, nitrogen and sulfur cycling (*Bhattarai et al., 2017*).

Trends in microbial community diversity were also used as indicators for influence of the PEI dredging on the reef sediments. Two discrete reef clusters which appeared in 2020 representing the south reef and north reef did not re-constitute in 2021 (Fig. 3). However, the changes observed in the south reef samples were most likely not the effects of dredging as the PEI sediments remained even more distinct from the reef sediments in 2021 and samples farther away from PEI, meant to serve as a control, also changed accordingly.

There was also no evidence that there was an increase in OTUs from Archaeal or Chloroflexi families families, which could potentially serve as indicators for the influence of dredging since these were observed to increase in ICW after dredging. This indicated that microbial diversity in reef sediments were not static. Thus, the reef microbiome dynamics were likely influenced by currents, tides, high volumes of ship, or boat traffic as previously suggested. The clustering of reef sediments in 2021 were due to higher abundances of cyanobacteria in 2020 (Fig. 4), suggesting light availability played a role. Resuspended sediment from any disturbance, dredging or boat traffic included, could cause shading that could decrease abundances of photosynthetic microorganisms, like cyanobacteria. This supports growing attention to resuspended sediments which are known to impact benthic reef or seagrass habitats (*Luter et al., 2021*).

It cannot be ruled out that dredging was responsible for the changes observed in microbial community structure of the south reef sediments in 2021 and reduction of cyanobacteria abundances. The control samples used to draw conclusions were assumed to be suitable controls due to their distance from the PEI and channel. However, the effects of dredging on sediment microbial communities could have extended farther south than anticipated. Dredging could have increased sediment resuspension and reduced light penetration to explain the decrease in cyanobacterial abundances. Future work to examine the range of influence on reef sediment in relation to proximity to the PEI should consider the possibility of farther-reaching impacts of dredging on sediment microbial communities. The disruption of biogeochemical processes that take place in reef sediments by the introduction of microbial communities or shading from PEI sediment could have substantial implications on coral reef ecosystem health.

Despite the maintenance dredging event that occurred, reef and PEI sediment microbial communities remained distinct in both years, highlighting the impact of "built environments" and human activity on natural environments. Specifically, we also found that these differences correlated with the increase in many trace metals and minerals (Table S2, Fig. S4). Most metals correlated to microbial diversity in the PEI and strongest to the diversity in the intracoastal, where there was the greatest influence from dredging based on changes in microbial diversity. Commercial ports and harbors are recognized sources of trace metal contamination in the environment and thus can serve as indicators of anthropogenic influence (*Di Giulio & Scanlon, 1985*; *Taylor, Birch & Links, 2004*; *Krahforst, Sherman & Kehm, 2022*, *Phillips et al., 1992*). While trace metals and Se were in higher concentrations in the PEI sediments regardless of dredging activity, Se increased in the reef sediments in 2021. This could also support influence from the PEI maintenance dredging. Regardless, trace metal contamination is both a public health and ecological concern, and release and re-distribution of trace metals and minerals due to dredging can have many negative impacts that expand beyond the health of the coral reefs and other nearby habitats.

### Detection of potential human and coral pathogens in PEI and reef sediments

The Florida Department of Environmental Protection (DEP) reported 2,779 wastewater spills in Florida in 2019, including a December 2019 burst in Fort Lauderdale that released 211.6 million gallons of raw sewage into canals and streets. Thus, close monitoring and determining the effects of urbanization, runoff, and subsequent water and sediment quality remain an ongoing concern for environmental protection managers and the wider public. There was no evidence that human pathogens increased in the PEI or reef sediments after maintenance dredging, however several were identified in the sediments at both locations. Again, this highlights the impact of human activities on natural ecosystems that ultimately extend to the coral reef habitat.

Since 2014, SCTLD has been decimating coral populations on the Florida reef tract in Broward to the Florida Keys (U.S.A, *Meiling et al., 2020*). Previous observations point to a Port of Miami (Florida, U.S.A) deepening project in 2013–2015 which was coincident with the rise of SCLTD (*Miller et al., 2016*; *Cunning et al., 2019*). *Bessell-Browne et al. (2017)* found that concomitant reduced light from turbidity was important, and subsequent microbiome studies point to the potential for SCTLD pathogen sources from the dredging activity.

*Rosales et al. (2020)* found relatively higher abundances of Rhodobacterales and Rhizobiales in SCTLD coral tissue, possibly pointing to a pathogen association. These taxa were also present in the PEI and reef sediments analyzed in the present study. *Becker et al. (2021)* more recently identified other bacteria that could be associated with SCTLD, such as *Cohaesibacter*, *Algicola*, *and Thalassobius*. These taxa can also be found in sediments of the PEI and reef but were at relatively much lower abundances compared to Rhizobiales and Rhodobacterales. Although there was no evidence that these taxa increased after dredging, this points to the sediments as a possible source for coral pathogens in this area (*Studivan et al., 2022*). To advance coral disease studies in the future, reef sediment microbiomes including viruses from healthier reefs at different or more remote Caribbean locales should be characterized in depth. This comparison could reveal which bacteria contribute to healthy *vs* deteriorated reef tracts (*Peixoto et al., 2017*). These data will provide a baseline for the types of bacteria which occur in sediments and may be resuspended in the water column after disturbances (*e.g.*, dredging, boat use, storms).

## CONCLUSIONS

This study using microbial diversity as an indicator to investigate the effects of dredging on a major port provides an interesting comparison of microbial diversity in a near century old human "built" environment (the PEI) adjacent to nearby "natural" and sensitive reef environment. The results from this study indicated that (i) microbial diversity between built and natural environments are distinct; (ii) these port *vs* reef differences remain mostly consistent over 1 year and (iii) microbial beta diversity was more dynamic within each area, shifting potentially due to anthropogenic influence (dredging). There is evidence of possible disturbances *via* the O&M dredging within the PEI, primarily in the ICW, and possibly the reef sediments identified by distinct changes in microbial diversity and an

increase in trace metal contamination. Although maintenance dredging did not increase the relative abundances of these taxa, evidence for the sediments being a potential source of both human and coral pathogens was also observed. This study highlighted the impact of human activity can drastically alter an environment and the possibility of impacts of surrounding natural environments. This is especially important to consider for environments that are sensitive to change, such as coral reefs and mangroves which are in very close proximity to many built and urban environments in South Florida, since their health is already degrading.

## ACKNOWLEDGEMENTS

We are also grateful to Colleen McMaken for assistance with R and PRIMER-E, Christian Rodriguez and Kristin Olenchak for valuable help with DNA extractions, Dr. Yan Ding from Florida International University for analysis of water and sediment samples, Paisley Samuel with GIS and map figures, and Alex Delgado's team at Industrial Diver Corporation for sample collections. We also thank Jocelyn Karazsia and Dr. Xaymara Serrano and Kristen Donofrio at US Army Corps of Engineers (USACE) for background information on PEI dredging activities.

### Funding

This project fulfilled Contract PO # B6F91C issued by the Florida Department of Environmental Protection (DEP) Coral Reef Conservation Program and funded by a grant agreement from the Florida Department of Environmental Protection (DEP) through National Oceanic and Atmospheric Administration (NOAA) Award No. NA19NOS4820053 to DEP. FDEP CRCP provided initial input for the research goals, which was carried out by the PI and team. The funders had no role in study design, data collection and analysis, decision to publish, or preparation of the manuscript.

### Grant Disclosures

The following grant information was disclosed by the authors:
Florida Department of Environmental Protection (DEP) Coral Reef Conservation Program.
National Oceanic and Atmospheric Administration (NOAA): NA19NOS4820053 to DEP. FDEP CRCP.

### Competing Interests

The authors declare that they have no competing interests.

### Author Contributions

- Lauren E. Krausfeldt performed the experiments, analyzed the data, prepared figures and/or tables, authored or reviewed drafts of the article, and approved the final draft.

- Jose Victor Lopez conceived and designed the experiments, analyzed the data, prepared figures and/or tables, authored or reviewed drafts of the article, and approved the final draft.
- Catherine Margaret Bilodeau performed the experiments, analyzed the data, prepared figures and/or tables, and approved the final draft.
- Hyo Won Lee performed the experiments, analyzed the data, prepared figures and/or tables, and approved the final draft.
- Shelby L. Casali conceived and designed the experiments, analyzed the data, authored or reviewed drafts of the article, and approved the final draft.

## Data Availability

The sequences are available at the National Center of Biotechnology Information (NCBI) Sequence Read Archive (SRA): PRJNA742832.

## Supplemental Information

Supplemental information for this article can be found online at http://dx.doi.org/10.7717/peerj.14288#supplemental-information.

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
