# Peer review of "Change and stasis of distinct sediment microbiomes across Port Everglades Inlet (PEI) and the adjacent coral reefs"

_PeerJ, doi:10.7717/peerj.14288_

## Round 0.1 · original submission · Major Revisions

I have received reports back from two expert reviewers and agree with their evaluation that this is an interesting study. Both reviewers have made a number of suggestions and comments that will improve the manuscript. Please respond to all of these in a revised version of the manuscript.

·

Basic reporting

Clear & professional English for sure. Not too many typos, but where they happen it interrupts the clarity of some sentences, so I have pointed the ones I saw out.
Good job on literature references.
Professional article structure, some information in figures & figure captions is missing.

Experimental design

Mostly no comment. Just need some more details on sample sequencing.

Validity of the findings

The statistics are lacking in some areas, or they need clarification where reported.
Lots of conclusions in discussion were not specific enough regarding taxa or didn't reference the parts of the results that supported their findings.

Additional comments

Summary:
The authors present a very interesting study where they thoroughly sampled sediment at a port and nearby reef before and after a dredging project. They conducted microbial community sequencing and found some differences in these communities compared between the port and reef samples. The introduction was especially informative. The methods, results, figures, and discussion need some more details and proofreading, especially in terms of reporting statistics and clarifying comparisons. I have provided comments to help clarify things for readers of the article (for example, me) below.

Abstract
L40: “relative ease” – I don’t know if many people new to the microbiology field would feel the same, I would re-phrase this

Introduction
L49 & 60: reporting it first in meters & then feet confused me, I had to convert to check if you were talking about the same depth
L74: ‘any ecosystem’ feels very absolute & rigid, what about just saying “valuable information about ecosystems” or “profiling reveals valuable ecosystem information”
L91: should be ‘provides’
L90-93: the last 2/3 of this sentence are very confusing, can you simplify please?

Materials & methods
L104: can you clarify your first usage of these abbreviations? TN, TP, TOC, TC. Will help more readers
L104-6: I’m not understanding what sample IDs you’re referring to. For instance, in figure 1 it says P02 etc., but there’s way more than a meter between P01 and P02 so that can’t be it.
L105-6: can you re-phrase this sentence about sampling in triplicate? Do you mean you took three samples within some kind of distance of each other from the top 10 cm at every location noted on Figure 1? At the very least, I would take out “Therefore” since it doesn’t follow from the previous sentence as far as I can tell, and “essentially” – which makes it sound unsure
I’m also confused about what you’re pointing out with the specific sites here at the end of this paragraph, as I’m not understanding how these site names correspond with the black & white dots on Figure 1. Please clarify.
L112: this is the first mention of mesocosms, to what are you referring to exactly?
L122: capitalize Bacteria and Archaea (I’m pretty sure all taxonomic names except species capitalized), check for this elsewhere in the manuscript as well
L119-125: were there any library prep steps to summarize between amplifying PCR product & sending for sequencing?
L127: cite Qiime2
Materials & methods generally: was a negative control sequenced?
Was there any filtering of data belonging to mitochondria, chloroplasts, etc.?
Were the 2020 & 2021 libraries sequenced together or separately?
Information on alpha diversity isn’t given here, but it’s important to clarify whether that analysis was done on square-root transformed data (which is what I understand from results is what the Bray-Curtis comparison was done on?) or on raw data. It’s the most conservative to do both normalized & raw data comparisons to make sure significant values aren’t an artefact of different count sizes.

Results
L148: Table S3 doesn’t appear in the supplementary materials that I downloaded
L149-151: I wrote this comment in the figure 2 caption below, but I would re-phrase “species richness” when referring to Chao1. This happens in a few other places besides this line as well, please check.
Also, there was no mention of conducting Chao1 analysis in the Methods, please fix.
L151-152: this is a Discussion sentence, not results
L157-9: it feels strange to mention here the results of the samples that served as control, but I don’t see where the actual Chao1 values were written or included in a figure for these controls. Please clarify with more specific values than “trends were not the same” for these samples.
Similar to last comment, I would remove instances of where you say “this comparison was different” & keep where you say the direction of the differences. For example, “sediment samples were more diverse than reef samples” is more informative than “there were differences between reef and sediment”. However, it makes sense to not have a direction of the differences in clustering.
Finally, stats were reported in the next paragraphs of results but not this one on diversity, please fix. Similarly, types of statistical tests run should have been mentioned in Methods as well.
L161: Bray-Curtis dissimilarity
L163: remove ‘the’
L163: I’m confused here which comparison this is. Is it comparing both 2020 & 2021 samples together & saying the statistic is significant between reef & port? Please clarify.
L166: I don’t think R should be negative?
Throughout manuscript, whether there’s a space between numbers & signs should stay consistent. For instance, this paragraph has both “p = 0.89” & “p=0.001”
Throughout this paragraph, keep putting “ANOSIM” before “R=..” or else specify if it was a different statistical test.
L167: put “p=” before 0.001
L167: I don’t understand again which comparison this is specifically. Please clarify.
L173-174: I don’t see what you mean about the ICW samples driving these differences in the S1 plot.
L176-77: are the control samples labeled in the NMDS again in figure S1?
L174-77: p-values?
L179: ‘was’ Proteobacteria
L181: add “classes” so all the different taxonomic levels don’t get confusing
L188-191: what metric did you use to calculate “average similarity” it’s not a phrase I’m familiar with
L189: I’m not familiar with “Desulfobubaceae” & I don’t see anything on google, can you check the spelling for me please?
Data on microbe abundance: no statistics or even numerical summary values are given when describing what is “more abundant.” There are many programs that specifically quantify which taxa are more statistically abundant, such as DESeq2 & ANCOM. If there are no stats, then saying even 6.6% vs. 4.6% in a given taxa between two variables would be helpful rather than just saying “it was higher”.
L218: reference Table S2 again if that’s where to find the pvalue.
L225: typos in this sentence

Discussion
L240: cite specific figures that support the statement
L241: say specifically some examples of what microorganisms you’re talking about here..
L243: high concentrations of what
L245-247: I have trouble with saying there was a higher diversity of cyanobacteria here if you didn’t pull out cyanobacteria from the data & specifically quantify that to support this statement. If you feel one of your figures supports this statement, then cite the figure here.
^I have the same statement as the previous one and L241 for other mentions of taxa throughout the discussion, for example those that are anaerobic. Not enough specifics. If you’re worried the new discussion will be too long, I would recommend making the discussion of the current speeds more concise as this was not included in your data.
L316: fix typo in 2015
L330: ‘these’ data instead
L331: typos in the middle of the sentence, please clarify
L332: I wouldn’t classify storms as a human disturbance
L346: change to ‘surrounding’
L347-349: there are typos in this sentence after ‘that are sensitive to change’
Make sure you have discussed all of your important results in the discussion, such as separately discussing implications of alpha & beta diversity differences. Essentially, paragraphs of discussion should touch on all of your 5 figures. This was hard to check since the Figure numbers were not referenced.

Figures
Figure 1 caption: if I’m following correctly, the white circles actually indicate sediment & water together, correct? Just says water in the figure caption.
Figure 1: do Atlantic Ocean & ICW need to be sideways?
Figure 2 caption & throughout: I wouldn’t call Chao1 just “species richness,” it is its own metric estimating species richness. I would just say Chao1 richness or Chao1 index. I could be wrong, happy to see a counterexample.
There’s a space between Chao & 1 in the second usage of it in the caption.
Figure 2: I feel like putting statistical groups “a” “ab” “c” etc. on this boxplot would be more clear, but I’m also okay with it as is too.
100% need to put ‘Chao1 index’ on the y-axis, can’t just leave unlabeled numbers. I would also say “Year & Habitat” or something under the x-axis labels, but no big deal if not, it’s pretty self-explanatory.
Figure 3 caption: it’s Bray-Curtis dissimilarity, not similarity.
Figure 3: I would put (a) 2020 and (b) 2021 at the top so readers can follow without even reading the caption. Similarly, you need to have a visual color legend of what the different color points mean for site on the actual plot itself instead of just in text in the figure caption.
Finally, this last one is less important than the other two, can the statistical ellipses be black & grey or something? Having them blue & green but also the different sites are blue & green is visually confusing.
Figure 4 caption (this also applies to Figures 1, 3, and 4 generally): which samples define subsections “south reef” “channel” etc. is not denoted anywhere that I can see (let me know if I missed it), since the IDs in Figure 1 are just R or P for reef or port.
Also confused about the phyla making up 95% of the total community, but y-axis goes up to 100%
Figure 4: put “Phyla” as legend title for (a) and put “Order” on y-axis for (b)

Figure S1 caption: There are lots of things different between Figure 3 and this one. Write a new caption for S1.
Figure S2: the heat bar needs a legend title, I assume ‘relative abundance’? Label which taxonomic level on the y-axis as well
Figure S4: what are the blue & grey shaded areas? Needs to be mentioned in legend & figure caption. Also what is the asterisk.

Reviewer 2 ·

Basic reporting

Lopez et al. present a study that investigates the potential influence of dredging on marine microbial communities and trace metal concentrations in sediments. The results suggest that dredging in the port affected microbial community composition in the nearby Intracoastal Waterway (ICW) but not reefs further from shore.

Overall, the article is well written but there are several sentences throughout that contain minor grammatical errors or omissions (eg, l. 276 "mostly likely" should be "most likely") and the authors should carefully proof-read their revision to fix these mistakes.

I would also like to encourage the authors to make reference to figures and tables throughout the discussion where appropriate to help the reader follow their arguments. It is at times difficult to follow the arguments being made. For example, the reader has to scan through multiple figures and supplements to find the clusters referenced on lines 264/265. Why not point the reader to the appropriate figure?

Experimental design

The authors were able to sample baseline samples and samples directly following the disturbance event (dredging), as they were aware of the planned activities. This allowed the authors to show that shifts in microbiome composition and trace metal concentrations changed in the port following dredging. It would be nice to collect samples again in the future to test if and when these values return to baseline.

One criticism of the sampling design, that the authors acknowledge, is that their control samples may not be true controls due to their close proximity to the reefs sampled. Here, samples from more distant reefs with less anthropogenic influence may have been a better choice. The authors acknowledge this in their discussion (lines 273-287). However, I find this part of the discussion hard to follow, as it relies on interpretation of figures that are not explicitly referenced. I would suggest revising this paragraph that highlights some of the limitations of the sampling scheme to make the argument the authors are trying to make clearer.

Validity of the findings

The authors conducted standard microbiome sequencing and collected trace metal data (I understand more about the former than the latter). The data analyses (barplots of taxonomic composition, nMDS, alpha diversity comparisons) seem appropriate and relevant to the question that is being addressed. An indicator species analysis could strengthen the part of the discussion that focuses on the authors' comparisons of taxon composition differences between port and reef. As is, the author seem to rely on the interpretation of taxon abundance barplots to identify taxa of interest that distinguish the different sampling time points and locations. Here, using an explicit test to identify the taxa that drive differences in community composition may be helpful to identify key drivers of change. Alternatively, these taxa could be identified from the loadings of their multivariate statistics.

One concern is the lack of detail on how OTUs were identified. I understand that the authors used a data analysis provider for this step. However, the authors should provide additional detail for this step of their analysis. Eg, at which similarity level were OTUs clustered? Generally, OTUs have fallen out of favor and amplicon sequence variants (ASV) are now the preferred method for identifying taxa in metabarcoding datasets. I do not think that using ASVs would change the big picture conclusions but the authors may want to address this in the M&Ms of the manuscript.

The term species is at times used synonymous with OTU. The authors should use OTU consistently or possibly taxon/taxa instead, considering that OTUs may not be resolved to the species level. Line, 155, for example, mentions species when OTUs would seem more appropriate.

Additional comments

Minor comment: check the manuscript for units of measurement. Line 60 lists water depths in feet; meter should be provided in addition.

---

## Round 0.2 · Minor Revisions

I am satisfied with the changes made to the manuscript,. However, I have pointed out some minor issues with redaction that I have pointed out in the PDF of the tracked changes version. For future reference, it makes the work of the editor far easier if your comments and addition within the text are references with line numbers in the rebuttal.

·

Basic reporting

Yes! Except for some mixup with Figures 4-6

Experimental design

All good!

Validity of the findings

All good!

Additional comments

This paper was thoroughly improved during revision, thanks for making the changes! A few small comments:

Methods:
L138 reads “are displayed using letters over the boxplots.” – in the .pdf I see, there are stars above the boxplots, so I would remove this line from the methods or replace ‘letters’ with ‘asterisks’ here.
Results:
L167-8: ‘undredged controls’ – these aren’t marked on Figure 2 so not sure why Figure 2 was referenced here, unless I missed something on the figure.
L238: Figures 4 & 5 look repeated, or I just can’t see what the difference is between them, and you reference Figure 5 here to be talking about metals so I think there’s an error in the figures.
Discussion:
L308: ‘in’ is missing after ‘decrease’

Reviewer 2 ·

Basic reporting

I appreciate the effort the authors made in providing clear references to figures and samples in both results and discussion sections in their revision. The original submission was somewhat hard to follow in places without these references. In my opinion, the revised manuscript is greatly improved and now easy to follow.

Experimental design

Potential shortcomings of the experimental design are now clearly delineated in the discussion. This was a concern I had in my earlier review that has been addressed.

Validity of the findings

The earlier version of the article lacked information on the methods used to analyze the metabarocding sequence data. This issue has now been fixed.

The authors now provide relative abundances of taxa of interest in the results and their relative abundances. In addition, where appropriate, statistical tests were used to determine if OTU abundance differences observed between samples were significant. These additions address the comment I made in my earlier review.

---

## Round 0.3 · accepted · Accept

I am satisfied with the changes made to the manuscript.